# Clustering-Based Component Fraction Estimation in Solid–Liquid Two-Phase Flow in Dredging Engineering

**DOI:** 10.3390/s20195697

**Published:** 2020-10-06

**Authors:** Chang Sun, Shihong Yue, Qi Li, Huaxiang Wang

**Affiliations:** School of Electrical Automation and Information Engineering, Tianjin University, Tianjin 300072, China; sun_chang@tju.edu.cn (C.S.); qili_2017@tju.edu.cn (Q.L.); hxwang@tju.edu.cn (H.W.)

**Keywords:** component fraction, solid–liquid two-phase flow, ERT, conductivity

## Abstract

Component fraction (CF) is one of the most important parameters in multiple-phase flow. Due to the complexity of the solid–liquid two-phase flow, the CF estimation remains unsolved both in scientific research and industrial application for a long time. Electrical resistance tomography (ERT) is an advanced type of conductivity detection technique due to its low-cost, fast-response, non-invasive, and non-radiation characteristics. However, when the existing ERT method is used to measure the CF value in solid–liquid two-phase flow in dredging engineering, there are at least three problems: (1) the dependence of reference distribution whose CF value is zero; (2) the size of the detected objects may be too small to be found by ERT; and (3) there is no efficient way to estimate the effect of artifacts in ERT. In this paper, we proposed a method based on the clustering technique, where a fast-fuzzy clustering algorithm is used to partition the ERT image to three clusters that respond to liquid, solid phases, and their mixtures and artifacts, respectively. The clustering algorithm does not need any reference distribution in the CF estimation. In the case of small solid objects or artifacts, the CF value remains effectively computed by prior information. To validate the new method, a group of typical CF estimations in dredging engineering were implemented. Results show that the new method can effectively overcome the limitations of the existing method, and can provide a practical and more accurate way for CF estimation.

## 1. Introduction

Solid–liquid two-phase flow is widely encountered in many process industries such as process control and pipe transmission in practice [1]. Flow component fraction (CF) estimation plays an important role in the process detection and parameter analysis of multiple-phase flow. The high precision of CF is necessarily required to effectively control industrial processes [2]. Despite CF being studied using various tomographic modalities such as a single source *γ*-ray computed tomography [3], ultrasonically-based detection techniques [4], positron emission particle tracking [5], etc., but as a valuably imaging technique, electrical resistance tomography (ERT) [6] provides both a cross-sectional image of the detected field and the CF value in solid–liquid two-phase flow [7]. In particular, compared with other tomographic techniques, ERT is an advanced and nondestructive technique with fast-response, non-invasive, and low-cost characteristics in obtaining 2D/3D distribution parameter information [8,9,10].

The ERT-based CF estimation methods have evolved for many years, and each progress provides information of better accuracy and stronger robustness [11,12]. Various estimation methods can be categorized for hardware reformulation and algorithm revision [13,14], and our research in this paper focused on the latter. Almost all CF estimation algorithms result from the Maxwell–Garnett (MG) formula [15], but the preliminary MG method has a limited application range due to some inevitable assumptions and conditions. Particularly, for the solid–liquid two-phase flow in dredging engineering [16], the MG method remains rather inaccurate due, at least, to the following three problems:(1)Each dispersed phase is assumedly even-distributed in practice and thus can be presented by the same conductivity value [17]. However, solid and liquid objects are dynamically changeable and it is difficult to satisfy the assumption.(2)The ERT reconstruction of the detected field has natural limitations such as the ill-posed problem and ‘soft field’ effect [18,19]. These limitations make the CF estimation have uncertain, inconsistent, and incomplete characteristics. However, the MG method cannot really reflect these natural characteristics in the detected field, and thus the accuracy of the estimated CF value is not guaranteed.(3)The ERT image is of low spatial resolution that may make the detected objects have inevitable and random artifacts, and in most cases, objects with small size are undistinguishable at all. Consequently, the estimated CF value by MG may greatly deviate from the real one in practice.

In the past, some studies have attempted to improve the existing CF estimation algorithms. For instance, numerical simulation results have been presented for the phase volume fraction of the solid–liquid multiphase flows in a horizontal curved pipe [20]. To raise measurable range of CF, various gas void fractions have been tested [21,22], and the void fraction was estimated based on the polynomial regression of measurement voltage values. Recent research [23] has shown that if the sensitivity matrix in the ERT process is carefully chosen, the CF estimation can be improved by integrating the prior information available in practice. More related reviews can be found in [24]. Furthermore, a class of interface-capturing methods [25,26] has attracted significant attention for simulating flows involving extensive topological changes. For example, an extension of the coupled volume-of-fluid and level-set method [27] was presented for simulating free surface flows in arbitrary 2D polygon meshes. The numerical results also indicated the abilities of the proposed method in handling a large density ratio and its adaptability to different grid types. The proposed method can be regarded as a promising approach for simulating free surface flows occurring in more complex geometries. However, these existing methods are not designed to solve the above problems essentially under the condition of dredging engineering, where the applicable method is necessarily non-parametric and assumption-free, otherwise the users cannot determine these parameters. Consequently, developing a feasible CF estimation method is an emergency task. 

Unlike the existing methods, in this paper, we propose a clustering-based CF estimation algorithm to solve the above problem by two steps. First, the fast fuzzy cluster algorithm (*f-*FCM) [28] was used to partition any ERT image to different clusters relative to solid and liquid phases. The use of *f-*FCM aims to find ERT-distinguishable objects and reduces the effect of noisy data and the “soft field” effect. Second, these ERT-undistinguishable objects were recomputed by prior information that are constructed in the proposed method. Finally, the two steps were combined to raise the CF estimation accuracy. The proposed method was compared with the typical MG method under actual experiments to recover the applicable range and the weaknesses of our proposed method.

## 2. Related Work

For any 2-D field, ERT reconstructs the detected objects by boundary measurements after alternatively injecting currents or voltages into one or several electrodes (see Figure 1a). When the detected flow objects enter the field (see Figure 1b), ERT can visually reconstruct the distribution of these objects and thereby compute flow parameter by the measurements available. 

To visually reconstruct the detected objects, the ERT field is discretized by rectangular or triangular units (pixels) (see Figure 1a). In the ERT imaging process, a detected field is partitioned into *n* units*/*pixels whose conductivity distribution are solved by using *m* measurements (voltages). According to the perturbation theory and linearity assumption [23], the conductivity distribution of *n* pixels obeys the following equation: *U* = *Sσ*(1)
where *U =* {*u_i_*} ∈ *R^m^**, σ =* {*σ_j_*} ∈ *R^n^* refer to measurements and conductivity distributions, respectively; *S* = {*s_ij_*} ∈ *R^m^*^×*n*^ is called the sensitivity matrix [29], where *s_ij_* represents the effect from the *j*th unit to *i*th measurement, *I* = 1, 2, …, *m*; *j* = 1, 2, …, *n*. The most widely used ERT imaging algorithm is the linear back projection (LBP) [30], which solves *σ* by using the fast and direct form as follows: *σ* = *S*^−1^*U*(2)
where *S^−^*^1^ is the inverse matrix of *S* and directly taken as the transposition of *S* in LBP, and the distributions of *σ* can be visually represented by various gray levels of *n* pixels and the conductivity of the *j*th pixel *σ_j_* is visually presented by its gray level, *j =* 1, 2, …, *n*, and denotes *G* = {*g_i_*} ∈ *R^m^.*

After the conductivity of each pixel is determined, the CF value of the dispersed phase can be computed. Various correlations have been proposed to solve CF, and the Maxwell–Garnett (MG) formula [17] is the most widely used CF estimation method with the following form:(3)$$\alpha =\frac{2\sigma_{1}+\sigma_{2}-2\sigma_{mc}-\sigma_{mc}\sigma_{2}/\sigma_{1}}{\sigma_{mc}-\sigma_{2}\sigma_{mc}/\sigma_{1}+2(\sigma_{1}-\sigma_{2})}$$
where *σ*_1_ is the conductivity of flowable and continuous phase (e.g., seawater); *σ*_2_ is the conductivity of the detected solid phase (e.g., soil or sand); and *σ**_mc_* is the conductivity mixed of the two phases. In the case of the non-conductive dispersed phase, *σ*_2_ is nearly 0 sm^−1^ and the MG formula is reduced as
(4)$$\alpha =(2-2\sigma_{mc}/\sigma_{1})/(\sigma_{mc}/\sigma_{1}+2)$$

Equation (4) shows that the conductivity ratio *σ**_mc_**/**σ*_1_ is the only variable to determine the value of *α*. Hence, the computational way is very realizable in practice. However, the MG method is not directly applicable since the mean of the dispersed phase must be computed. In practice, liquids, solids, and their mixture cannot clearly be distinguished in advance. To overcome the problem, the sensitivity matrix *S* and pixel conductivity/grey distributions *G* have been used for the CF computation. As a typical example, Ding et al. [31] reformates the MG method by the following equation:(5)$$\alpha =-2(\bar{g}/a_{{}_{T}})/(\bar{g}/a_{T}+3),\;s.t.,a_{T}=\sum_{i=1}^{m}S^{T}U$$
where $\bar{g}$ is the mean of the grey levels of all pixels after obtaining grey distributions *G*. Thereby, the applicable range of Equation (4) can be widened. However, in most cases, the mean cannot effectively reflect the real grey distribution, so the estimated CF by ERT remains rather inaccurate. 

In this paper, we used the clustering technique for the CF estimation process in dredging engineering. To overcome the problem of the preliminary MG method, three changes were made as follows:
(1)The *f*-FCM algorithm was used for the CF estimation process in the solid–liquid two-phase flow. The use of fuzzy clustering, rather than other clustering algorithms, aims to overcome uncertainty, uncompletedness, and inconsistency in the ERT imaging process.(2)All detected objects were categorized into distinguishable and undistinguishable sets by ERT, respectively. These distinguishable objects can be estimated by *f*-FCM, whereas undistinguishable objects are computed by prior information. In this paper, the prior information was perfectly determined and represented by an inquiring table that was constructed in advance.

The proposed method was designed to compute the CF value generally and effectively in dredging engineering, and is helpful in overcoming the limitations of the existing method.

## 3. Cluster-Based Component Fraction (CF) Estimation Method

The detected objects in solid–liquid two-phase flow can be categorized into distinguishable and undistinguishable types by ERT, and accordingly, our proposed method is illustrated by two cases.

### 3.1. Computation on Distinguishable Objects by the Fast Fuzzy Cluster Algorithm (f-FCM) Algorithm

If all detected objects in a field have a large enough size, these objects can be distinguished in an ERT image. Figure 2a,d simulates two types of solid–liquid objects, (b) and (e) are their respective ERT images, and (c) and (f) are their statistical histograms related to all grey distributions of the pixels, where each bar along with the horizontal and vertical axis refer to the grey level interval and pixel number, respectively. These solid-phase and liquid-phase objects in (a) have a clear boundary, and these solid objects (in black) can be found in the related ERT image, as shown in (b). Any ERT-based CF method can correctly estimate the CF value with small error. However, when partial solid objects (e.g., two small circles in (a)) are dissolved or mixed into liquid-phase objects (see (d)), the ERT image cannot find them (see (e)) and the CF estimation may generate a large error. In particular, the statistical histogram (c) is more difficult to partition to two clusters than (f), since the latter has a more centralized peak whereas two separated peaks relative to solid and liquid objects are desired. Furthermore, any ERT image inevitably contains artifacts that cannot correctly be estimated by the existing method when computing the CF value. Consequently, the pixels that respond to the mixtures and artifacts make the real CF value estimation difficult to correctly estimate. 

To solve the above problems, we first used the *f*-FCM algorithm to partition all pixels in any ERT image into three clusters: solid, liquid, and their mixtures or artifacts. In the *f*-FCM algorithm, all *n* pixels are projected into *L* grades of grey levels and represented as the discrete set:*A* = {1, 2,…, *L*}(6)

Let *H*(*l*) be the number of pixels at the *l-*th grey level, *l* = 1, 2,…, *L,* then the objective function of the *f*-FCM algorithm is
(7)$$J_{m}=\sum_{l=1}^{L}\sum_{i=1}^{c}{\left(u_{il}\right)}^{m}\cdot H(l)\cdot {\left(l-v_{i}\right)}^{2}$$
where *u_il_* is the membership degree of the *l*th grey level to *i*th cluster, *I* = 1, 2, 3; *c* is taken as 3, which refers to the partitioned number of clusters; and *m* is a weighting exponent, and is often taken as 1.5, if no other information is available. *f*-FCM updates cluster center and membership function as follows: (8)$$v_{i}{}^{\left(k\right)}=\sum_{l=1}^{L}{\left(u_{il}{}^{\left(k\right)}\right)}^{m}\cdot H(l)\cdot l/\sum_{l=1}^{L}{\left(u_{il}{}^{\left(k\right)}\right)}^{m},\;i=1,\;2,\;3$$
where *k* is iteration times, and
(9)$$u_{il}{}^{\left(k+1\right)}={\left[\sum_{i=1}^{c}{\left(\frac{l-v_{i}}{l-v_{l}}\right)}^{1/(m-1)}\right]}^{-1},\;i=\;1,\;2,...,\;c;\;l=\;1,\;2,...,\;L$$

The algorithm stops if the total residual error ||∑(*u_il_*^(*k*+1)^ − *u_il_*
^(*k*)^)|| is smaller than a specified value *ε*, or *k* reaches a maximum number of iterations. In this paper, the termination residual error and the maximum number of iterations were taken as 10^−5^ and 100, respectively.

Assume that all pixels in *G* in an ERT image is partitioned by the *f*-FCM algorithm to three clusters *C*_1_, *C*_2_, and *C*_3_ that have respective centers *v*_1_, *v*_2_, and *v*_3_, and respective numbers of pixels *n*_1_, *n*_2_, and *n*_3_ (see Figure 3). Let *D*_1_(*j*) and *D*_2_(*j*) be the distance of the *j*th pixel in *C*_3_ to clustering prototypes *v*_1_ and *v*_2_, respectively. Following common practice, first, the values of *D*_1_(*j*) and *D*_2_(*j*) reflect the respective possibility that the *j*th pixel belongs to *C*_1_ and *C*_2_. Closer distance to a center a pixel has, and larger possibility the pixel belongs to the cluster. Second, these pixels in *C*_3_ result from artifacts or mixtures in *C*_1_ and *C*_2_, and the ratio that these pixels in *C*_3_ belong to *C*_1_ and *C*_2_ is assumed to be directly proportional to the values of *n*_1_ and *n*_2_, respectively. 

According to the following twin-weighted rule, all pixels in *C*_3_ are partitioned to *C*_1_ and *C*_2_,
(10)$$num(1)=\frac{w_{1}}{w_{1}+w_{2}}n_{3}\;and\;num(2)=\frac{w_{2}}{w_{1}+w_{2}}n_{3},\;s.t.,w_{1}=r_{1}n_{1},\;w_{2}=r_{2}n_{2}$$
where *num*(1) and *num*(2) are the number of pixels that are individually assigned to *C*_1_ and *C*_2_, and
(11)$$\left\{\begin{array}{l} r_{1}={\left(1+D_{1}(k)\right)}^{-1},\;k=1,2,\ldots ,n_{3};\\ r_{2}={\left(1+D_{2}(k)\right)}^{-1},\;k=1,2,\ldots ,n_{3}; \end{array}\right.\;$$

The number of 1 in Equation (11) aims to prevent the value of *r*_1_ and *r*_2_ being overlarge. Consequently, the CF values in the solid and liquid phases are individually estimated as
(12)$$CF(1)=\frac{n_{1}+num(1)}{n}\;and\;CF(2)=\frac{n_{2}+num(2)}{n}\;$$

Table 1 shows a group of ERT images by LBP, and evaluation results of *CF*(1) by the MG method, where *CF*(1) with respect to the solid component fraction is taken from 10% to 40% increasingly, and the conductivity of objects and backgrounds are set to 0.01 and 10, respectively. Here the bars in red, green, and blue in these histograms refer to the three partitioned clusters by *f-*FCM, respectively.

From these various models, the obtained accuracy by Equation (12) was higher than the MG method in all four models, whereas the MG method tends to underestimate the real value in most cases. When the ERT image contains artifacts, MG causes a large error, but Equation (12) is affected to a small extent. These results show the correction of Equation (12) when the detected solid-phase objects can be distinguished by ERT. Therefore, the CF estimation by Equation (12) does not need other prior information, and thus is easily realized and comprehensive. 

### 3.2. Computation on Undistinguishable Objects by Prior Information Inquiry

If any solid object size is very small, ERT cannot find the object at all, so Equation (12) must be inaccurate. Let five simulating solid–liquid flow patterns have the same value of *CF*(1), 0.360, but different solid object sizes. Table 2 shows the ERT images, partitioned clusters by *f*-FCM, and computed values of *CF*(1) by Equation (12) on the five patterns, respectively. As solid object size gradually decreases so that ERT cannot distinguish most solid objects, when using *f*-FCM, these undistinguishable objects are incorrectly partitioned to *C*_2_ or *C*_3_ and make the estimated *CF*(1) generate a very large error. Therefore, *CF* (1) in Equation (12) must be corrected and recomputed. 

Note that these pixels in *C*_1_ are very different from those in *C*_2_ or *C*_3_, and the former basically consists of solid-phase objects whereas the latter may contain partial solid-phase objects that ERT cannot distinguish. Hence, we estimated *CF*(1) by two steps. First, all pixels in the ERT image are clustered by *f*-FCM to construct a set that is still denoted as *C*_1_ with the number of pixels *n*_1_, which responds to these distinguishable solid-phase objects by ERT. Then, all undistinguishable objects that are contained in the rest of the pixels are estimated by a grey index: (13)$$\sigma (C_{u})=\sum_{g_{i}\in \{\Phi -C_{1}\}}^{}g_{i}/n=\sum_{g_{i}\in \{\Phi -C_{1}\}}^{}s_{ij}u_{i}/n$$
where both values of *s_ij_* and *u_i_* are available in any ERT process, and Φ consists of all pixels in the detected field with the number of pixels *n*. The number ‘*n*’ in Equation (13) aims to normalize the value of *σ*(*C_u_*) to each pixel. Generally, in dredging engineering, the conductivity of liquid-phase pixels (e.g., seawater) are known in advance. Furthermore, assuming that actual solid-phase objects in Φ consist of *n*_s_ pixels, if *C*_1_ is determined by *f*-FCM and *n*_s_ is fixed, (*n_s_–n*_1_) is just the number of pixels that construct undistinguishable objects. Generally, the value of *σ*(*C_u_*) is nearly unchangeable regardless of how the (*n_s_–n*_1_) pixels are distributed, as explained below. 

Table 3 shows a group of randomly distributed pixels that respond to undistinguishable objects, where the values of *n_s_*/*n* are always taken as 0.40, but the value of *n*_1_/*n* in all models was selected as nearly 0.30. Therefore, the value of *σ*(*C_u_*) is approximately unchangeable. In fact, according to the Ohm theorem [21], the area and the length of the method from injecting to measuring electrodes is nearly unchangeable under various pixel distributions if the value of (*n*_1_–*n_s_*) is fixed. The conclusion was verified by our previous work [28]. Consequently, according to the value of *σ*(*C_u_*) and the determined pixels in *C*_1_ by *f*-FCM, the interrelation between (*n_s_–n*_1_) and *σ*(*C_u_*) can be determined prior. 

Figure 4a shows the varied curve of *σ*(*C_u_*) as the two relative variables of *n*_1_ and (*n_s_–n*_1_) are changed. It can be observed that (*n_s_–n*_1_) is tightly relative to the value of *σ*(*C_u_*). Alternatively, Figure 4b shows the changing trend of (*n_s_–n*_1_) as *σ*(*C_u_*) increases when *n*_1_/*n* = 0.3.

Table 4 shows the correlation between *n*_1_, *σ*(*C_u_*), and (*n_s_–n*_1_). It provides a way to estimate the value of (*n_s_–n*_1_) when *n*_1_ and *σ*(*C_u_*) are known. Consequently, after determining the value of *n*_1_ by *f*-FCM and the value of (*n_s_–n*_1_) by *σ*(*C_u_*), the value of *CF*(1) can be computed as: *CF*(1) = *n_s_*/*n*(14)

According to Equation (14), the proposed CF estimation method is presented in Algorithm 1:
**Algorithm 1.** The proposed CF estimation method.**Input:** Boundary measurements and sensitivity matrix *S*
**Output**: *CF*(1)(1)Image the detected field by ERT;(2)Cluster all pixels by *f*-FCM;(3)Determine *C*_1_ and *n*_1_;(4)Compute *σ*(*C_u_*) by Equation (13);(5)Inquiry (*n_s_–n*_1_) by prior knowledge based on *σ*(*C_u_*);(6)Compute *CF*(1) by Equation (14).

Hereafter, we call the cluster-based CF estimation CBCF.

## 4. Experiment

The experiments were implemented in the TianShan dredge in the dredging process. The dredged solids and seawater were mixed as a solid–liquid two-phase flow in a mixing tank, and then further transferred into a pipeline. The ERT sensor was equipped in the transformation pipe with a diameter of 750 mm, as shown in Figure 5, and at the same position, an existing dredge-equipped Co-60 ray-based meter measured the cross-sectional CF in real time and provided a comparable CF value for the ERT sensor with various CF estimation methods. Note that the existing sensor is environmentally polluting and harmful to users, so the ERT system aims to replace it, but keeping the same measuring accuracy. The CF estimation aims to determine the solid-phase objects in the solid–liquid flow in which solid objects (soil, sand, stone, etc.) have low conductivity of about 1.0 S·m^−^^1^ and seawater has high conductivity of about 35 S·m^−1^ (see Figure 5b). Hence, their conductivity differences are very large and suitable for the measurements of any ERT system. The measuring data from the sensor are transmitted to the ERT system and are used to reconstruct the detected object distribution. CBCF and MG with Equation (5) were used to estimate the CF value from the ERT image and test their accuracy and efficiency. 

The CF value of solid-phase objects were adjustable by means of transverse and rotating velocities of the cutterhead in thee dredge. The inlet liquid velocity was 8 m/s, and the CF value ranged from about 0.30–0.40. The measured flow was basically in turbulence. Using a 16-electrode twin-plane ERT sensor, 208 independent measurements were obtained for a single frame of ERT image. The sampling rate of the used ERT system was 330 frames per second, and the reconstructed images was 120 frames per second by using the LBP algorithm. Computations were carried out using a PC with a CPU of Pentium^®^ 4 2.93 GHz and 2 GB RAM. An exploratory evaluation involved a group of typical CFs by adjusting transverse and rotating velocities of the cutterhead in the dredging process. Table 5 shows the clustered ERT images by the *f*-FCM algorithm, where the second and the fourth rows respond to two cases when the two-phase flow basically contains distinguishable and undistinguishable objects by ERT, respectively, and each column has nearly the same CF value collected from all available measurements.

With a visual look at these clustered images in the second row, these ERT-distinguishable solid- and liquid-objects can construct the clustering structure. Consequently, the area of distinguishable solid-phase objects in the dark is directly proportional to the relative CF values, but these objects in the fourth cannot be observed due to the existence of many undistinguishable objects. 

Table 6 shows the computed CF values individually by CBCF, *f*-FCM with Equation (12), and MG in comparison to the real value from the Co-60 ray-based meter. The ten CF values were the means of computed CF values in 10 continuous sampling phases, respectively, and the use of *f*-FCM with Equation (12) aimed to show the working process of CBCF.

Table 6 shows that both MG and CBCF can effectively estimate the CF value for the former five patterns in the second row in Table 5, since most objects can be distinguished by ERT. They have a relatively small error. However, for cases in the fourth row, these solid particles in Table 5 had too small a size to be distinguished by ERT and MG tended to underestimate the real CF whereas CBCF had a small error. In fact, the *f*-FCM with Equation (12), which is the first step of CBCF, also underestimated the real CF, but the second step of CBCF based on Table 4 counteracted the underestimation. 

Alternatively, a continuous sampling process with 1000 samples in the dredging process was tested at a velocity range from low to high and CF value ranges in [5%, 15%], and most solid-phase objects had a very small size so that *f*-FCM could not distinguish them. Moreover, the MF method has difficulty in correctly estimating the CF value due to the asymmetric and uneven distributions of all objects, whereas CBCF in fact estimates the CF value in two steps: first finding the objects with a large size by *f*-FCM, then inquiring the rest of the objects by the value of *σ*(*C_u_*) prior. Figure 6 further shows the accuracy of the estimated velocity by CBCF and MG compared with the real CF value. 

It can be clearly seen that the estimated CF curve from CBCF was very close to the real curve, whereas MG always underestimated the real values of CF since most undistinguishable objects by ERT could not be found by MG either. These results demonstrate that CBCF can outperform MG in the underlying solid–liquid two-phase flow in dredging engineering. 

However, CBCF needs an additional time cost for the clustering process, and in the averaging sense, the additional time has nearly 376.5% times as many as MG in the tested experiments. In fact, the existing MG method does not need any clustering process, however, the additional running time in CBCF is not unacceptable in dredging engineering.

## 5. Conclusions

The conventional CF estimation method based on the ERT image is problematic in solid–liquid two-phase flow measurement since the inevitable three limitations may often be encountered. The ERT system itself has difficulty in overcoming these natural limitations. Our proposed cluster-based method has the ability to decrease the negative effect of these limitations. The preliminary results presented in this paper show that the proposed method is comprehensive and effective for estimating the CF values. When all detected objects are categorized to ERT-distinguishable and ERT-undistinguishable types, the proposed CF estimation method has the potential to be applied to the CF estimation in most solid–liquid two-phase flow. The used evaluation formula and inquiring method for the CF estimation is easily realized in practice. 

However, the proposed method in this paper remains incomplete when facing various flowing conditions and arbitrarily-distributed detected objects. Moreover, the ERT imaging process has natural limitations, which must be overcome by additional prior information. Hence, it is necessary to construct a prior complete set to the CF estimation by the sum of grey levels of all pixels. In addition, the used *f*-FCM leaves the room for further reformulation.

## Figures and Tables

**Figure 1 sensors-20-05697-f001:**
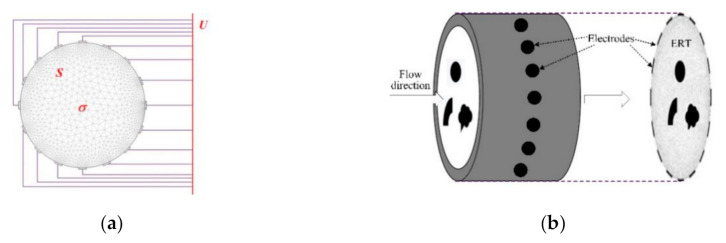
Electrical resistance tomography (ERT) sensor and measurements process. (**a**) ERT electrode plane and detected field. (**b**) Discretized ERT field.

**Figure 2 sensors-20-05697-f002:**
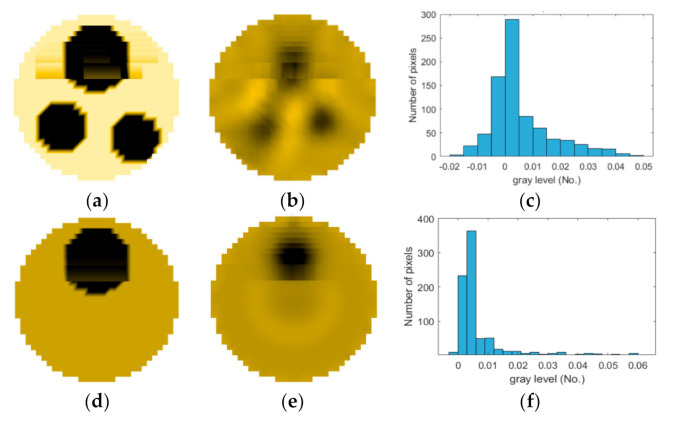
Comparison of two imaging conditions. (**a**) Separated solid and liquid objects. (**b**) ERT image of (**a**). (**c**) Statistical histogram of (**b**). (**d**) Mixed solid and liquid objects. (**e**) ERT image of (**d**). (**f**) Statistical histogram of (**e**).

**Figure 3 sensors-20-05697-f003:**
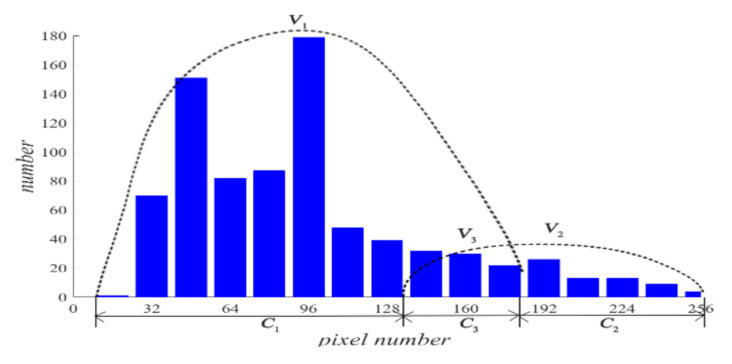
The partitioned three clusters by the *f*-FCM algorithm.

**Figure 4 sensors-20-05697-f004:**
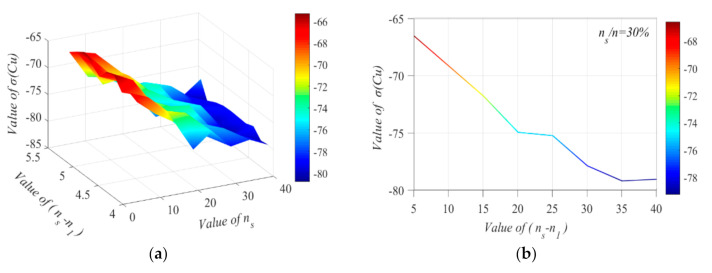
The varied curve of *σ*(*C_u_*) on two relative variables of *n*_1_ and (*n_s_–n*_1_). (**a**) *σ*(*C_u_*) under various(*n_s_–n*_1_) and *n*_1_. (**b**) Relation between *σ*(*C_u_*) and(*n_s_–n*_1_).

**Figure 5 sensors-20-05697-f005:**
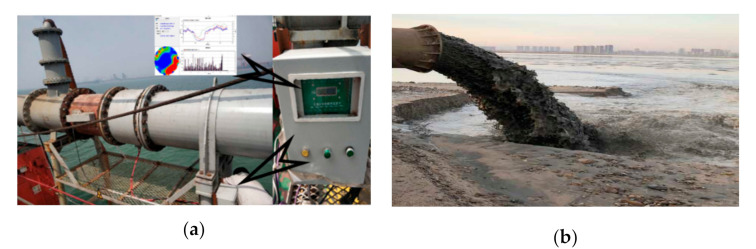
Experimental facility in dredging engineering. (**a**) ERT sensor and measuring meter. (**b**) Solid–liquid flow in pipe.

**Figure 6 sensors-20-05697-f006:**
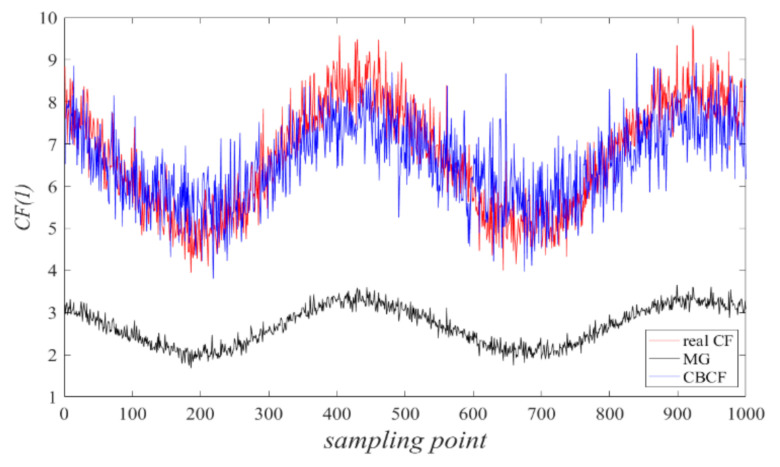
The estimated values of CF by CBCF and MG.

**Table 1 sensors-20-05697-t001:** Estimated component fraction (CF) values by Equations (2) and (12).

CF	Model	ERT	Statistical Histogram	MG	Equation (12)
10%	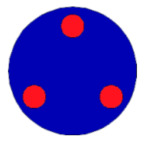	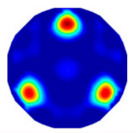	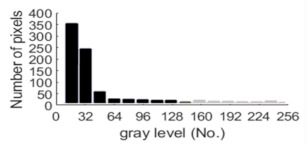	14.16%	8.74%
20%	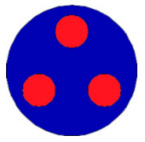	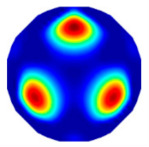	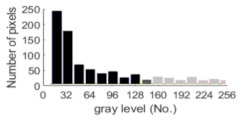	18.57%	20.44%
30%	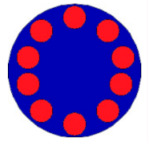	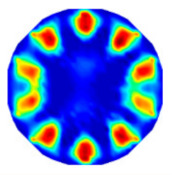	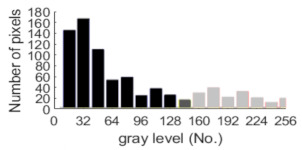	27.65%	29.56%
Legend:  background  artifacts  object

**Table 2 sensors-20-05697-t002:** The electrical resistance tomography (ERT) image, partitioned clusters, and computed *CF*(1) of five simulating flow patterns.

Computed CF(1)	0.358	0.366	0.310	0.225	0.096
ERT image	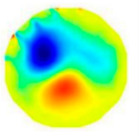	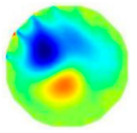	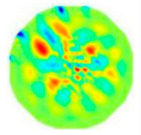	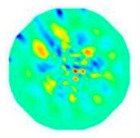	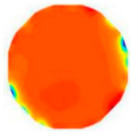
Partitioned clusters	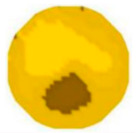	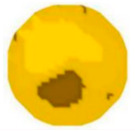	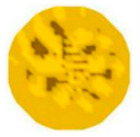	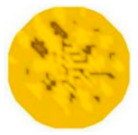	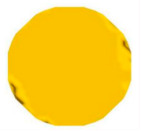

**Table 3 sensors-20-05697-t003:** Unchangeable value of *σ*(*C_u_*) at *n*_1_*/n* = 0.30 under various object distributions.

Pattern	Distribution	σ(Cu)	Pattern	Distribution	σ(Cu)
Model 1	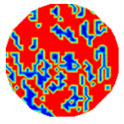	−78.8	Model 6	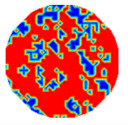	−79.5
Model 2	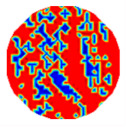	−76.4	Model 7	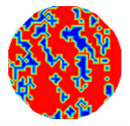	−78.9
Model 3	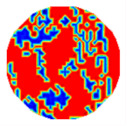	−78.4	Model 8	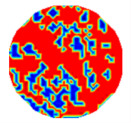	−79
Model 4	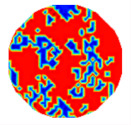	−78.9	Model 9	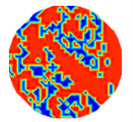	−76
Model 5	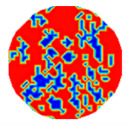	−75.4	Model 10	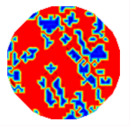	−78.1

**Table 4 sensors-20-05697-t004:** Inquiry value on (*n_s_–n*_1_) based on known *n*_1_ and *σ*(*C_u_*).

Model		1	2	3	4	5	6	7	8	9	10
5%	*σ*(*Φ-y*)	−67.2	−66.6	−67.6	−65.5	−65.8	−67.8	−66.5	−65.3	−66.3	−67.4
	*n*_1_-*n_s_*	3	0	3	2	0	3	3	2	0	3
10%	*σ*(*Φ-y*)	−70.1	−68	−68.7	−67	−67.9	−69.4	−69.1	−72	−67.2	−72.7
	*n*_1_-*n_s_*	44	44	44	43	44	45	42	47	44	48
15%	*σ*(*Φ-y*)	−72.3	−71.3	−72.5	−72.3	−72.7	−72.8	−70	−70.5	−71.4	−73.5
	*n*_1_-*n_s_*	84	84	86	87	87	86	81	84	83	86
20%	*σ*(*Φ-y*)	−78.1	−72.3	−75.7	−73.2	−75.2	−77.2	−73.2	−75	−74.3	−75.9
	*n*_1_-*n_s_*	125	126	125	125	126	125	125	125	127	127
25%	*σ*(*Φ-y*)	−75.3	−77.2	−75.8	−73.6	−75.7	−74.8	−76	−74.1	−74.8	−75.7
	*n*_1_-*n_s_*	167	169	166	165	166	167	166	165	165	167
30%	*σ*(*Φ-y*)	−78.8	−76.4	−78.4	−78.9	−75.4	−79.5	−78.9	−79	−76	−78.1
	*n*_1_-*n_s_*	208	208	209	206	207	206	206	207	206	206
35%	*σ*(*Φ-y*)	−78.2	−78.6	−79.4	−80.5	−79.7	−80	−78.6	−79.3	−78.7	−79.7
	*n*_1_-*n_s_*	247	247	249	247	247	247	248	247	247	247
40%	*σ*(*Φ-y*)	−79.9	−80.1	−78.5	−79	−78.9	−78.8	−78.9	−79.3	−80.8	−77.1
	*n*_1_-*n_s_*	288	289	287	287	287	287	288	288	289	287

**Table 5 sensors-20-05697-t005:** A group of CF estimations under various particle sizes.

**Real CF value**	**0.305**	**0.332**	**0.342**	**0.367**	**0.381**	**0.406**
Identifiable object by ERT	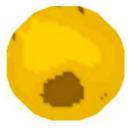	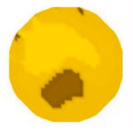	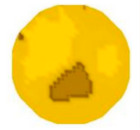	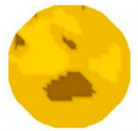	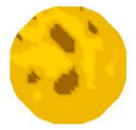	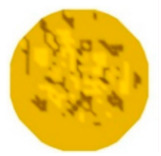
**Real CF value**	**0.307**	**0.330**	**0.341**	**0.370**	**0.388**	**0.401**
Unidentifiable object by ERT	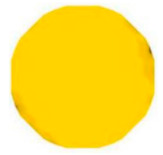	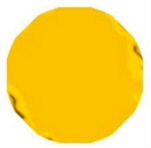	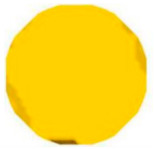	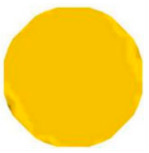	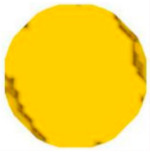	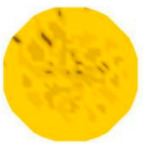

**Table 6 sensors-20-05697-t006:** The computed CF values by CBCF, *f*-FCM with Equation (12), and MG compared with real CF.

Real CF	7.00	5.28	5.48	7.52	8.24	6.86	5.19	5.66	7.58	8.13
*f-*FCM	2.96	2.73	2.72	3.01	3.02	2.90	2.92	2.81	2.90	2.95
MG	5.79	4.17	4.23	6.97	7.24	2.73	2.16	2.31	3.03	3.24
CBCF	6.66	5.75	5.76	6.99	7.44	6.60	5.89	5.97	7.10	7.48

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
