# Peer review of "Clustering-Based Component Fraction Estimation in Solid–Liquid Two-Phase Flow in Dredging Engineering"

_sensors, 2020, doi:10.3390/s20195697_

Round 1

Reviewer 1 Report

The article raises the following doubts in terms of legibility, relevance and science:

  1. The descriptions of the axes in Figures 2b and 2d are illegible. It can be concluded from the description that it is about gray levels, but the description of the axes themselves and their explanation in the text should be more prescriptive, so as not to leave room for guesswork.
  2. Similarly in Figure 3.2. There is a visible histogram (third from left) which has axes and legends completely unreadable. I don't know what blue, green and red mean.
  3. The problem of unreadable axes and descriptions also applies to Figures 1b, 4a, 4b, and 6.
  4. Attempts to simulate various multiphase flows have resulted in a range of different numerical approaches. Among them, implicit interface-capturing methods such as Volume of Fluid and Level-Set have attracted significant attention for simulating flows involving extensive topological changes. Why do the authors validate the presented method (CBCF) only with the Maxwell-Garnett formula (MG)?
  5. In the research, the authors use fuzzy cluster algorithm to partition any ERT image to different 65 clusters relative to solid and liquid phases. However, it is known that there are many other algorithms that can be used for this purpose. So why did the authors use this and not another? I believe that this should be discussed and described in much more detail in the paper. So far, the use of fuzzy cluster algorithm is not convincing. Even if the authors' approach is new, it must be proved that it is better than the approaches used so far. A comparison only with the Maxwell-Garnett formula is not enough.
  6. In the Introduction one should clearly (in a separate paragraph) define the purpose of the work and the novelties. The authors are expected to provide an overview of the currently used methods along with a description of their advantages and disadvantages. Against their background, you can present your own concept, listing its advantages, but not forgetting its weaknesses.

Author Response

First, we would like to thank all the reviewers for their new insightful comments and valuable suggestions. This manuscript has been carefully revised according to the reviewers’ constructive comments and suggestions. All changes have been marked in red in the revised mc. The detailed responses for the comments and suggestions from the reviewers are as follows.

Responses for the comments from Reviewer #1:

  1. The descriptions of the axes in Figures 2b and 2d are illegible. It can be concluded from the description that it is about gray levels, but the description of the axes themselves and their explanation in the text should be more prescriptive, so as not to leave room for guesswork.

Reply:  Thankful for the reviewer pointing out these drawbacks, which have been improved in this revised mc.

  1. Similarly in Figure 3.2. There is a visible histogram (third from left) which has axes and legends completely unreadable. I don't know what blue, green and red mean.

Reply:  These figures have been further explained by adding illustration of related colors and legends.

  1. The problem of unreadable axes and descriptions also applies to Figures 1b, 4a, 4b, and 6.

Reply:   These figures have been revised. Thanks.

  1. Attempts to simulate various multiphase flows have resulted in a range of different numerical approaches. Among them, implicit interface-capturing methods such as Volume of Fluid and Level-Set have attracted significant attention for simulating flows involving extensive topological changes. Why do the authors validate the presented method (CBCF) only with the Maxwell-Garnett formula (MG)?

Reply:Thankful for the reviewer’s insights. These progresses that the reviewer mentioned are very important achievements in the research field of multiphase flows. But as far as the solid-liquid two-phase flow is, the applicable method necessarily is non-parametric and assumption-free, otherwise the users cannot determine these parameters. Nevertheless, these interface-capturing methods are constructive to our researches, we have again evaluated the advantages and disadvantages of our proposed method and added related analysis in the revised mc.

  1. In the research, the authors use fuzzy cluster algorithm to partition any ERT image to different 65 clusters relative to solid and liquid phases. However, it is known that there are many other algorithms that can be used for this purpose. So why did the authors use this and not another? I believe that this should be discussed and described in much more detail in the paper. So far, the use of fuzzy cluster algorithm is not convincing.

Reply:  Thankful for the reviewer’s professional comments. Indeed, there are many clustering algorithms that can cluster ERT image, but the fuzzy clustering has been the most major image segmentation algorithm so far owing to strong ability of handling uncertain information and easy performance. Specially, the fast fuzzy clustering method (FCM) that is used in this paper can turn the task of clustering a large number of pixels to a group of grey grades with very limited number. Therefore, it is helpful to raise the time resolution of image segmentation.

        Also, we agree the reviewer’s comments very much. The previous mc fails clearly to express the reason of using the fast FCM algorithm. Now they have been added in the revised version.

  1. Even if the authors' approach is new, it must be proved that it is better than the approaches used so far. A comparison only with the Maxwell-Garnett formula is not enough.

Reply:  Yes. We have added the explanations and illustrations to verify the reasons of using the fast FCM algorithm. Also, a group of comparable experimental results has added to verify the merits of our proposed method. However, some typical existing method to CF estimation cannot be used to compare with our proposed method, since they aren’t designed to compute the solid-liquid two-phase flow in the dredging engineering.

  1. In the Introduction one should clearly (in a separate paragraph) define the purpose of the work and the novelties. The authors are expected to provide an overview of the currently used methods along with a description of their advantages and disadvantages. Against their background, you can present your own concept, listing its advantages, but not forgetting its weaknesses.

Reply:  Thankful for the reviewer’s suggestions very much. In a separate paragraph, the applicable range of the new method have been reviewed in the introduction parts, and whereby its advantage and disadvantages have further been illustrated.

Reviewer 2 Report

The paper presents interesting method of CF estimation in a multi-phase flow. In general the paprer needs correction of English grammar and some spelling errors.
Some detailed remarks:
1. Abstract: ERT is not a new technique.
2. Resolution of figures and charts is not acceptable, it is too low. In print they are almost unreadable, even magnified PDF is difficult to read.
3. Some references are printed bold, some use normal font. It should be corrected.
4. Line 119, should be Figure 2
5. line 159, text printed in blue,
6. line 163, It is not clear what is the meaning of CF(1), please clarify it.
7. Table 2, please clarify the statement, that CF(1) is the same around 0,36, but in the table CF(1) decreases from 0,358 to 0,096
8. Table 3, row 1 needs reformatting,
9. Figure 4, NG value is not explained,
10. line 211, Table 3, it should be Table 4, similarly table 4 to 5.
11. Table 3/4 column x-y needs reformatting

Calculations and simulations were performed for conductivities 1 and 35 Sm-1. Results for conductivities, which differs less would be interesting and will allow to derive more general conclusions regarding applicability of the proposed method.

Author Response

First, we would like to thank all the reviewers for their new insightful comments and valuable suggestions. This manuscript has been carefully revised according to the reviewers’ constructive comments and suggestions. All changes have been marked in red in the revised mc. The detailed responses for the comments and suggestions from the reviewers are as follows.

  1. Abstract: ERT is not a new technique.

Reply: It has been revised as “…advanced…”. Thanks.

  1. Resolution of figures and charts is not acceptable, it is too low. In print they are almost unreadable, even magnified PDF is difficult to read.

Reply: All figures and charts have been examined, redrawn and revised in the new mc.

  1. Some references are printed bold, some use normal font. They have been corrected.

Reply: Sorry for these careless. They have revised in the revised mc. 

  1. Line 119, should be Figure 2

Reply: Indeed, it is a typing error. All indexes in the paragraph should be Figure 2, and have been revised.

  1. line 159, text printed in blue,

Reply: Sorry. It has been revised. 

  1. line 163, It is not clear what is the meaning of CF(1), please clarify it.

Reply: Sorry for the unclear expression, and we have further added explanations in the new mc. 

  1. Table 2, please clarify the statement, that CF(1) is the same around 0, 36, but in the table CF(1) decreases from 0, 358 to 0, 096

Reply: Sorry for these inaccurate expressions, and we have further clarified it in the new mc.

  1. Table 3, row 1 needs reformatting,

Reply: Yes, it has been reformatted in the revised mc. 

  1. Figure 4, NG value is not explained,

Reply: Sorry for the error. It has been corrected.

  1. line 211, Table 3, it should be Table 4, similarly table 4 to 5.

Reply:  All of them have been revised.

  1. Table 3/4 column x-y needs reformatting

Reply: Sorry for the typing error, it has been reformatted in the revised mc. Thanks.

Reviewer 3 Report

Comments and Suggestions for Authors

This paper is devoted to solve an important problem of Electrical Resistance Tomography (ERT) systems efficiency increasing for two-phase flows investigation for dredging engineering. To solve this problem the new method of component fraction (CF) estimation in two-phase flows (CBCF) developed by authors is considered and experimentally studied. The main feature of the proposed CBCF method is an original cluster-based approach and corresponding technique.

The introduction section of the paper proposes a wide enough review on recent methods, techniques and studies of ERT systems for two-phase flows analysis with 18 corresponding references. The fundamentals of proposed method and experimental data are detail considered and discussed. Comparison of the proposed CBCF method with conventional ERT method is performed.

The reviewer suggests that this paper will be very useful for specialists in the field of high-resolution ERT systems. This paper can be accepted after following minor corrections.

  1. On the line 85 the matrix S-1 is incorrectly represented as the transposition of the matrix S. In fact, in this case, S-1 is the inverse matrix of S. This term should be corrected.
  2. On line 93, the unit of electrical conductivity was written as “sm−1”. It should be rewritten as S×m–1.
  3. According to the text on lines 119-121, horizontal axes on the statistical histograms refer to grey level range. But the horizontal axis on the statistical histogram in Figure 3 is labeled “pixel number”, which is incorrect. Corresponding corrections should also be made to the text explaining Figure 3.
  4. According to the text on lines 194-196, “Table 3 shows a group of randomly distributed pixels that respond to undistinguishable objects… at n1/n nearly at 30%”. But the heading of the Table 3 defines (n1-ns)/n=30%. Which of these two expressions is incorrect? Please clarify.
  5. What is the abbreviation NG on the line 208 (the title of Figure 4)? The title of Figure 4 does not quite correspond to its content. Clarify, please.
  6. On line 207 it is written “(ns-n1) under various σ(Cu) and n1”. But in fact, Figure 4(a) shows σ(Cu) under various (ns-n1) and ns (not n1 according to the axis title). Please clarify.
  7. Table 3 is not formatted correctly.
  8. On line 214 “…method is present as follow” should be rewritten as “…method is presented as follow”.
  9. The title of Table 5 is “A group of CF estimations under various particular sizes”. May be you mean “A group of CF estimations under various particle sizes”?
  10. On line 252 should probably refer to Table 6, not Table 5.
  11. The title of Table 6 is “The averaging accuracy of CF estimations by CBCF and MC(%)”. Please explain how the data in Table 6 were obtained. If these are the values ​​of the CF factor in %, then the Table 6 heading should reflect this. If these are the values ​​of the relative error in determining the CF factor in % by the methods under consideration, then the error of the proposed CBCP method turns out to be larger than that of the MG.
  12. The title of Figure 6 is “The accuracy of estimated velocity by CBCF and MG”. Probably, Figure 6 reflects CF(1) values in % for different methods? Clarify please.

Author Response

First, we would like to thank all the reviewers for their new insightful comments and valuable suggestions. This manuscript has been carefully revised according to the reviewers’ constructive comments and suggestions. All changes have been marked in red in the revised mc. The detailed responses for the comments and suggestions from the reviewers are as follows.

  1. On the line 85 the matrix S-1 is incorrectly represented as the transposition of the matrix S. In fact, in this case, S-1 is the inverse matrix of S. This term should be corrected.

Reply: Yes, the previous expression is confusing and has been corrected by adding an illustrating sentence.

  1. On line 93, the unit of electrical conductivity was written as “sm−1”. It should be rewritten as S×m–1.

Reply: Yes, it has been corrected as “s/m”. 

  1. According to the text on lines 119-121, horizontal axes on the statistical histograms refer to grey level range. But the horizontal axis on the statistical histogram in Figure 3 is labeled “pixel number”, which is incorrect. Corresponding corrections should also be made to the text explaining Figure 3.

Reply:  The previous figure lacks the correct description, and now has been revised in the new mc. Thanks. 

  1. According to the text on lines 194-196, “Table 3 shows a group of randomly distributed pixels that respond to undistinguishable objects… at n1/n nearly at 30%”. But the heading of the Table 3 defines (n1-ns)/n=30%. Which of these two expressions is incorrect? Please clarify.

Reply: Thankful for the reviewer’s careful reading. We have made them consistency by correcting the previous illustrations. Also, some necessary explanations have been added. 

  1. What is the abbreviation NG on the line 208 (the title of Figure 4)? The title of Figure 4 does not quite correspond to its content. Clarify, please.

Reply: Indeed, it is error. We have modified the title and explained the abbreviation.

  1. On line 207 it is written “(ns-n1) under various σ(Cu) and n1”. But in fact, Figure 4(a) shows σ(Cu) under various (ns-n1) and ns (not n1 according to the axis title). Please clarify.

Reply: Corrections have made in the revised mc, thanks.

  1. Table 3 is not formatted correctly.

Reply: It has been revised.

  1. On line 214 “…method is present as follow” should be rewritten as “…method is presented as follow”.

Reply: It has been corrected in the revised mc. Thanks.

  1. The title of Table 5 is “A group of CF estimations under various particular sizes”. May be you mean “A group of CF estimations under various particle sizes”?

Reply: sorry for the confusion. The title has been revised.

  1. On line 252 should probably refer to Table 6, not Table 5.

Reply: Yes, it has been revised. 

  1. The title of Table 6 is “The averaging accuracy of CF estimations by CBCF and MC(%)”. Please explain how the data in Table 6 were obtained. If these are the values of the CF factor in %, then the Table 6 heading should reflect this. If these are the values of the relative error in determining the CF factor in % by the methods under consideration, then the error of the proposed CBCP method turns out to be larger than that of the MG.

Reply: Sorry for these unclear descriptions. Some necessary illustrations have been added, and these meaning of these evaluations have been explained in the new mc. Thankful for the reviewer’s careful reading. 

  1. The title of Figure 6 is “The accuracy of estimated velocity by CBCF and MG”. Probably, Figure 6 reflects CF(1) values in % for different methods? Clarify please.

Reply:  Sorry for the error and it should be “CF”. We have corrected them in the revised mc.

Round 2

Reviewer 1 Report

I have no comments.